# Dimensions of Urban Blight in Emerging Southern Cities: A Case Study of Accra-Ghana

**Sally Adofowaa Mireku** [1,2,*], **Zaid Abubakari** [3] and **Javier Martinez** [2]

1   AnL Valuation and Property Consult, Kanda, Accra P.O. Box YK 627, Ghana
2   Department of Urban Planning and Management, Faculty of Geo-Information Science and Earth Observation (ITC), University of Twente, P.O. Box 217, 7500 AE Enschede, The Netherlands; j.a.martinez@utwente.nl
3   Faculty of Planning and Land Management, SDD University of Business and Integrated Development Studies, Wa P.O. Box WA64, Ghana; abubakari.zaid@gmail.com
*   Correspondence: sallimireku@gmail.com

**Abstract:** Urban blight functions inversely to city development and often leads to cities' deterioration in terms of physical beauty and functionality. While the underlying causes of urban blight in the context of the global north are mainly known in the literature to be population loss, economic decline, deindustrialisation and suburbanisation, there is a research gap regarding the root causes of urban blight in the global south, specifically in prime areas. Given the differences in the property rights regimes and economic growth trajectories between the global north and south, the underlying reasons for urban blight cannot be assumed to be the same. This study, thus, employed a qualitative method and case study approach to ascertain in-depth contextual reasons and effects for urban blight in a prime area, East Legon, Accra-Ghana. Beyond economic reasons, the study found that socio-cultural practices of landholding and land transfer in Ghana play an essential role in how blighted properties emerge. In the quest to preserve cultural heritage/identity, successors of old family houses (the ancestral roots) do their best to stay in them without selling or redeveloping them. The findings highlight the less obvious but relevant functions that blighted properties play in the city core at the micro level of individual families in fostering social cohesion and alleviating the need to pay higher rents. Thus, in the global south, we conclude that there is a need to pay attention to the less obvious roles that so-called blighted properties perform and to move beyond the default negative perception that blighted properties are entirely problematic.

**Keywords:** urban blight; prime area; customary land tenure; socio-cultural values; Sub-Saharan Africa; Ghana

## 1. Introduction

In recent decades, due to the increasing urbanisation and economic development, the demand for urban properties for various purposes, such as residential, commercial and industrial, has increased steadily. However, this increasing demand is hardly uniform across the city [1]. Prime urban areas are characterised by high property values, quality neighbourhoods and modernised developments. Nevertheless, some prime areas simultaneously experience urban blight in the form of vacant plots of land, abandoned structures, littered sites and degraded buildings, leading to the deterioration of cities in terms of physical beauty and functionality [2,3]. Urban blight is described by Weaver [4] as underinvestment in real property. This incidence of blight in the city presents an interesting dilemma where properties in prime areas, despite having great potential that can support useful developments, lie underused.

The phenomenon of urban blight dates back from cities in the global north, especially the United States of America (USA), United Kingdom (UK), and Germany, among others, after the industrial revolution era in the 19th–20th centuries. The incidence of the industrial

revolution in the 19th century caused an urban population boom where people migrated to the urban cities due to employment in manufacturing industries [5,6]. The subsequent economic decline resulted in urban cities becoming economically vulnerable, with industries collapsing, businesses decreasing and people losing their jobs [6]. Ultimately, these cities experienced physical and functional stagnation as real property, mainly of a residential and commercial type, became obsolete, with high vacancy ratios and a decline in maintenance [3]. Although several studies have investigated the main causes of urban blight in the global north to be population loss, deindustrialisation, economic decline and sub-urbanisation, few of such studies exist in the global south, predominantly in Sub-Saharan Africa (SSA). In SSA, land holdings and use practices are not separate entities from people and their belief systems but are constitutive and embedded in customary land tenure systems [7,8].

Different terminologies are used in describing cities' deterioration. For this study, the nuances of urban blight; urban decline, shrinkage and decay have been fostered into a mutual communication notion, "urban blight". This is because urban blight is a known concept in Ghana's urban land use policy. The current Land Use and Spatial Planning Act 2016 (Act 925) provides detailed descriptions of the criteria and the roles of the District Assemblies in tackling urban blight [9].

Specifically in Accra, Ghana's capital city, Appiahene-Gyamfi [10] argues that the city is knotted with cultural values; familistic, social lifestyles; and modernisation. While some stakeholders/actors attribute economic value to prime areas, others perceive urban spaces/properties differently, which do not match the modernised area [11,12]. Apparently, these unmatched properties, viewed as blight, are unevenly distributed within prime areas. While there is the possibility to leverage these blighted properties for the provision of modern housing or commercial development, some remain in the same conditions for many years for unknown reasons. Thus, the main question posed by this study is "*how can the existence of distributed pockets of urban blight in a prime area in Accra-Ghana be explained?*". The specific objectives are to determine the distribution of blighted properties in East Legon and ascertain the reasons from key stakeholders. This paper is structured as follows: Section 2 delves into the theoretical underpinnings and nuances of urban blight, as well as the perception of values attached to urban spaces. Section 3 describes the methodology. The results are presented in Section 4 and discussed in Section 5. Conclusions are drawn in Section 6.

## 2. Urban Blight and Value Systems

### 2.1. The Phenomenon of Urban Blight

The genesis of urban blight can be traced to the United States of America (USA). According to Gordon [3], use of the term occurred as early as 1918 in Philadelphia, where a planner described blight as an unbefitting district. This notwithstanding, different states in the USA fashioned their urban blight description based on peculiarities or uniqueness in their jurisdiction. For instance, Missouri state pronounced urban blight as overcrowding, inadequate light, ventilation and lack of sanitary facilities in an area, whereas New Jersey described urban blight to be abandoned industrial use; substandard, unsafe properties; and vacant lots. Additionally, California state added defective designs, either interior or exterior, to their urban blight description [3].

In the United Kingdom (UK), it was revealed by Haase et al. [13] that urban blight occurred in major commercial and industrial hub cities, such as Newcastle, Manchester, Liverpool, Birmingham and Glasgow. These cities experienced a population decrease and economic decline caused by the collapse of well-patronised commercial activities and auxiliary industries. Subsequently, there was suburbanisation, leading to the abandonment of industrialised cities. Additionally, Germany's case was likened to that of the United Kingdom (UK), which also evolved in the 1980s [14,15]. There was out-migration and deindustrialisation, resulting in population changes in cities. Nevertheless, in other parts of Europe such as Poland and Romania, the cause of urban blight was quite different.

According to Haase et al [13], urban blight in Poland and Romania resulted from a decline in the natural population change, high death rates and ageing of the population in the country.

From the global south perspective, however, existing literature reveals that in Latin American cities—Sao Paulo in Brazil and Guadalajara in Mexico—the causes of urban blight were similar to the global north. It was reported by Audirac et al. [16] that the causes of urban blight in these areas result from suburbanisation, deindustrialisation, and population loss. Similarly, in Africa, the most prominent cause of urban blight was the suburbanisation in South Africa in 1994. This was caused by a complex racial structure where a formerly white neighbourhood, Hillbrow, experienced an abandonment of houses by white people. Subsequently, the area was occupied by immigrants with low-income status who could not maintain the high standard of the area [17,18]. Furthermore, a study conducted by Reckien and Martinez-Fernandez [14] presented social factors to be the driving force for cities' blight in the Sub-Saharan African (SSA) region. The social factors given were hunger and epidemics like Human Immunodeficiency Virus Infection and Acquired Immunodeficiency Syndrome (HIV/AIDS). Nevertheless, no thorough explanations of the social factors were classified in their study. Many urban cities have issues relating to land use and development in Sub-Saharan Africa [19], yet little knowledge exists regarding the emergence of urban blight. Urban blight is likened to the accelerated growth of urbanisation, which is primarily caused by a high birth rate in urban cities and rural-urban migration. Notably, most of these urban cities were previously indigenous settlements [20]. Although urbanisation is good and has resulted in modern land use and development in African cities, rapid urbanisation, on the other hand, has resulted in unsustainable development where high population growth is not matching the existing urban infrastructure, especially housing [21,22]. Subsequently, the inadequate housing infrastructure has triggered the development of illegal settlements, leading to poor neighbourhoods that lack social amenities like water, toilet facilities and garbage bins, among others [22]. Neighbourhoods that lack basic infrastructure and are in disorder are also regarded as urban blight [3,23,24].

In Ghana, the enforcement of land use policies and laws is tackled by the local government. However, according to Cobbinah and Aboagye [20], the local governments do not have complete control over the enforcement due to the role played by traditional authorities in regulating and managing customary lands. They further explained that there is an inadequate collaboration between local governments and traditional authorities. Additionally, the current Land Use and Spatial Planning Act 2016 (Act 925) specifies the criteria for District Assemblies determining blighted properties, irrespective of the land tenure system, as either customary or statutory. Therefore, these criteria, stipulated in Section 103 of Act 925, set the basis and measures for identifying blighted properties in this study. They include:

a. "Irregularity of plots or parcels,
b. Inadequacy of street in the vicinity,
c. Lack of access to plots or habitable dwelling within the area,
d. Diversity of existing use which makes development control difficult or impossible,
e. Incompatibility with:

    i. The existing or proposed use
    ii. The spatial development framework and
    iii. The structure or local plan,

f. Adverse impact on the environment,
g. Overcrowding leading to unhealthy population density,
h. Lack of sanitation, drainage or appropriate service,
i. High incidence of crime which has been confirmed to be attributable to the type of development and
j. Safety or restriction to other authorised users"

As characterised by Act 925 [9], several commentators have also categorised urban blight from different perspectives. Such characterisation efforts enable a more global understanding of what urban blight could be, as well as the dynamics of its manifestation and nature. These are presented in the following section.

### 2.2. The Nuances of Urban Blight

There is a large body of literature on the deterioration of urban cities with diverse terminologies. The terminologies used in describing cities' deterioration differ from place to place, including the descriptions used by urban scholars such as city shrinkage, urban decay, urban decline, brownfields, or urban blight. However, whilst, on the one hand, Reckien and Martinez-Fernandez [14] assert that these terminologies may mean the same thing with regards to cities' physical characteristics and functioning, Haase et al. [25], on the other hand, argue that the emphasis and concepts of these terminologies are developed in diverse contexts, times, theoretical frameworks, and empirical backgrounds. Often, deterioration is studied at different geographical levels, either at the city or neighbourhood level. Urban blight is described in terms of real properties and/or urban spaces. Specifically, real properties consider the land and the buildings, while urban spaces are related to entire neighbourhood or city levels. According to Albers [26] in the history of urban planning, the changing attitudes of the population, which usually means the perspectives and priorities ranging between neglect and attention, affect the urban fabric, its beauty and landscape. Notably, this manifests in both the global north and south. In the global north, after the industrial revolution era, businesses declined and people lost their jobs. Subsequently, neighbourhoods became less attractive, experiencing physical and functional stagnation. Residences and commercial buildings have become obsolete, with high vacancy ratios and less maintenance culture by landlords due to low profits [1]. Livingston et al. [27] argue that there is a lack of inadequate effectiveness from local governments in satisfying the needs of such deteriorated neighbourhoods. Despite the level of deterioration, some residents may remain in the area as a result of low income. In the global south, on the other hand, Getis [21] argues that the deterioration of cities is partly due to rapid urbanisation. Inadequate housing infrastructure has triggered the development of unauthorised and illegal structures leading to poor neighbourhoods that lack basic social amenities such as water and toilet facilities [28]. Consequently, urban blight deteriorates cities' beauty and landscape [29,30]. The nuances of blight regarding different conditions, physical states, and uses and developments in urban settings, which are deemed contextual, are illustrated in Table 1.

**Table 1.** A summary of the nuances of urban blight.

| The Nuances of Urban Blight | Description & Sources from Literature |
| --- | --- |
| An idea with regards to the use of real property | Urban blight is described as an idea in the minds of various stakeholders concerning the condition, use and function of real property [21,22]. |
| The lack of basic urban infrastructure | Urban blight is an element that is caused by a lack of infrastructure [3,22]. |
| Neighbourhood disorder/lack of physical beauty | Many abandoned and deteriorated buildings in the area [23–25]. |
| Results in physical stagnation | The attributes of urban blight are visually demeaning and aesthetically depressing. This could lead to stagnation of land use and development in an area [30]. |
| A contributing factor to slum | Urban blight is an element that results in a slum [3]. Additionally, Breger [2] emphasises that historically, slums were regarded as blighted areas. |

**Table 1.** *Cont.*

| The Nuances of Urban Blight | Description & Sources from Literature |
| --- | --- |
| Comparable to urban decline/shrinkage | Weaver and Bagchi-Sen [31], Miekley [32] and Hoekveld [33] believe that the leading causes of urban blight from the global north perspective, such as poverty, unemployment, and vacancy, align with urban factors for decline/shrinkage. |
| The initial stage of urban decay | The severe phase of urban blight is used to describe urban decay's commencement [2]. Urban decay as explained by Fabiyi [34] is the neglect of the built environment symbolised by poor urban dwellers unable to repair their old structures. |

Authors' construct (2019).

Urban transformation programmes such as urban renewal and regeneration have been developed to reduce urban blight [3]. In the 21st century, however, the Sustainable Development Goals (SDG) framework has been established by the United Nations to guide developmental efforts between the years 2015 to 2030 [35]. According to De Vries and Voß [36], a greater percentage of the Sustainable Development Goals (SDG) is related to urban land, yet contemporary land management practices are fraught with issues of varied value systems. The level of utilisation of urban spaces thus differs regarding values, perceptions, priorities and reasoning [12]. The next section tackles the perception of values (economic, social and cultural) attached to urban spaces. While economic and social values are common in both the global north and south, in the global south, especially Sub-Saharan Africa (SSA), socio-cultural values are embedded in the land tenure systems, which makes the property rights regime unique and different from that of the global north.

### 2.3. The Perception of Values Attached to Urban Spaces

Values attached to urban spaces are discussed in this study because the issues of urban blight may be influenced by varying priorities and perceptions by stakeholders/actors. According to Galster [37], the four main actors who make use of an area are households, property owners, business holders and the local government. The households use the neighbourhood through the occupation of residential units. Additionally, the surrounding environments, like recreational facilities, add some form of residential satisfaction and quality to the use of the neighbourhood. Business holders, on the other hand, occupy non-residential facilities yet obtain some monetary value in the form of profit. As well as this, property owners occupy residential properties themselves or rent real property. Lastly, local governments mainly consume areas through tax revenues and provision of social amenities.

The explanations of the diverse values of the global north and south are described as follows: predominantly in the global north, the primary values attached to real properties are economic and social. Economic value is mainly associated with urban investment, where three of the aforementioned stakeholders of an area—business holders, local government and some of the property owners—make certain financial gains from the area. Somerville et al [38] emphasise that the economic structure of a country, real estate market, policies, and the level and nature of public goods and services determine the economic fabric of an area. Additionally, Galster [37] highlights that most of the residential neighbourhoods in the global north are established through large-scale construction. Nonetheless, the changes that occur afterwards are a result of how stakeholders attach value to the area. Hidalgo and Bernardo [39] therefore argue that the type and level of attachment placed on neighbourhoods and real properties differ in degrees and dimensions. When the level of social values exceed that of economic values, then the attractiveness of the neighbourhood and needs satisfaction are relatively assessed in comparison with other neighbourhoods by financially inclined stakeholders [40]. Ultimately, the decisions taken by wealthy actors affect the economic growth and development of the area, as well as the provision of public resources and services like recreational facilities by the local government. Others who

normally remain in deteriorated areas are those who attach social values to their urban spaces [41]. According to Scannell and Gifford [42], there are varied explanations for place and social attachment due to the cross-cutting nature of the notion in the fields of psychology, urban studies and environmental studies. In urban studies, however, place attachments are regarded as a strong emotional bond and sense of place that a group or an individual may have concerning a neighbourhood or a real property [41,42]. The social values, as inferred by researchers, are emotional bonds and affections developed over time in an area that results in strong networks and cohesion [42–44]. Additionally, Livingston et al. [41] and Johnston [44] agree that social value is a collective attachment to a place by homogenous people with common backgrounds. Instances of community attachment to places in the global north are native settlements such as Maori in New Zealand, Aboriginals in Australia, and Canadian and American Indians. These people believe their spaces are imbued with the spirit of their ancestors (spiritual identity), thus the need for heritage conservation. While the aforementioned assertion is not different for the global south, specifically in Sub-Saharan Africa (SSA), customary land tenure systems play a significant role, making up about ninety percent (90%) of landholdings in the region. As already mentioned, land holdings and use practices are not a separate entity from people and their belief systems but are constitutive and embedded in customary land tenure systems [7,8].

The conception of land in Sub-Saharan Africa (SSA) is nuanced and transcends the physical land per se. As described by Elias [45], land in Africa belongs to the living, the dead and the unborn. Sometimes, land is seen as a deity and entity from which people derive spiritual identity. A study by Abubakari et al. [8] in the Upper East region of Ghana explicates such spiritual connections with land where the earth priest (Tendaana) pacifies and sanctifies land allocations and transfers. Under customary tenure, the right to use land in a particular manner is contingent on one's gender, birth order and position within the social group [8]. Essentially, land is not disconnected from people, but the two are conflated and hardly separable. The transcendence in the conception of land makes customary law and practices reflect the exigencies of specific communities, although there are commonalities as well [30,31]. Thus, customary practices are not a coherent set of stable rules that apply uniformly across communities but evolve within and vary across communities [7]. From a system perspective, the embodiment of customary rules can be likened to a complex adaptive system whereby the actors (members) and the system (customary system) itself evolve, adapt and shape each other in a constantly evolving manner [46–48]. Customary lands are not only characterised by a communality in the manner they are held, but they are also characterised by an evolved set of norms and practices. Such norms and practices define membership and associated rights of land use, restrictions and responsibilities. Members usually have usufructuary interests (superior rights) and are given portions for their usage and livelihoods such as farming but sometimes without rights of disposition [49]. Although urbanisation, modernisation and economic development have resulted in increasing demand for properties in urban areas, such trends are still somewhat influenced by the resilient customary rules and belief systems. Landowners within cities still hold on to their beliefs and practices because of the continuity and preservation of their culture. According to Arko-Adjei [46], customary land tenure systems are usually unwritten, yet they are passed from one generation to another. In sum, the relationship between people with respect to land is governed by a continuously evolving set of rules, which are known within the social groups in which they are practiced. Distinct from statutory rules of tenure, customary rules of tenure are neither written nor consciously formulated at one point in time and are normally enforced within local circles. In the case of Ghana, where the constitution recognises customary law, this provides room for forum shopping and strategic choice making on the part of members [50].

## 3. Methodology

This study uses a qualitative research strategy coupled with a case study approach because the study aims to understand the contextual reasons for urban blight in Sub-Saharan Africa (SSA) in the case of Ghana (see Appendix A for analytical framework). According to Yin [51], the niche of a case study approach is to investigate a contemporary phenomenon that is labelled as "the case" in the real world. This research approach enables in-depth understanding of the pertinent contextual circumstances associated with the case. The choice of Ghana makes the study an interesting one to be investigated in SSA. This is because Ghana is the first SSA country to ascertain independence from colonisation in 1957 and thus the premier country to gain full control of urban planning and land management by an aboriginal government [52]. Additionally, the concept of urban blight is known in Ghana's urban policy, Land Use and Spatial Planning Act 2016 (Act 925), as previously mentioned. The research will provide a contextual understanding of urban blight to assist city planners and government agencies in operationalising urban policies.

### 3.1. The Study Area

This study is conducted in East Legon, a first-class suburb of Accra. The area measures approximately 4.95 square kilometres and falls under the Ayawaso West Municipal Assembly (AWMA), as shown in Figure 1. It is classified as a first-class area by the Assembly mainly due to good infrastructural facilities, such as roads, water, and electricity, inclusive of modern developments and major commercial activities. Additionally, it is recognised as one of Accra's most expensive areas, with an influx of international businesses and expatriates [53]. Despite being noted as an affluent area with a high demand for urban space, the area is experiencing blight, and some areas have been in the same condition for many years [54]. To understand the nature of urban blight in East Legon, the study was conducted in phases. These are described in the next section.

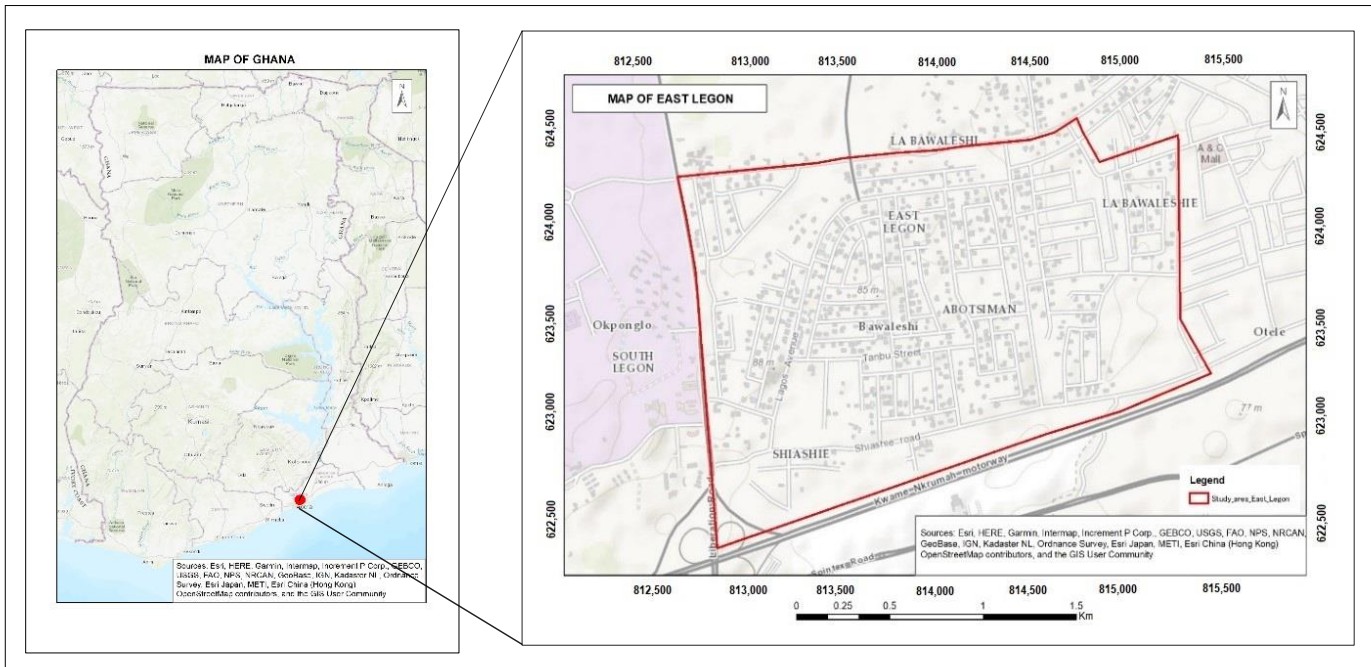

**Figure 1.** Map of East Legon, Accra-Ghana. Source: Esri Topographical Map and Land Use and Spatial Planning Authority.

### 3.2. Criteria for Identifying Blighted Properties

To identify blighted properties within East Legon, we used a two-stage criteria. The first stage entails definition and characterisation of urban blight, which paves a way for the second stage, a visual selection process based on the outcomes of the first stage.

In the first stage, we traced the characteristics of urban blight from the existing local law: Land Use and Spatial Planning Act 2016 (Act 925), as cited earlier in Section 2.1. Characterising urban blight according to the laws of Ghana is necessary to understand the case of Accra as a typical emerging southern city. A global characterisation would otherwise not match the local characteristics and thus have the tendency to blur or misrepresent the local reality/context of how urban blight is understood. We, however, acknowledge that a comparison between local and global perspectives is imperative to position our study within the discourse on urban blight. Thus, based on the characteristics of urban blight stipulated in Section 103 of the Land Use and Spatial Planning Act 2016 (Act 925), we categorised the characteristics into four forms of urban blight according to common descriptions found in literature [23,32,55]. These categorised forms of blight are shown in Table 2.

**Table 2.** A summary of the criteria for selecting blighted properties in this study.

| No. | Categorised Forms of Blight | Criteria According to the Land Use and Spatial Planning Act 2016 (Act 925) |
|---|---|---|
| (a) | Cluster of disordered settlements | "Irregularity of plots or parcels" |
|  |  | "Lack of access to plots or habitable dwelling within the area" |
|  |  | "Safety or restriction to the other authorised users" |
|  |  | "Overcrowding leading to unhealthy population density" |
|  |  | "Lack of sanitation, drainage or appropriate service" |
| (b) | Vacant plot/undeveloped land | "Safety or restriction to the other authorised users" |
|  |  | "Adverse impact on the environment" |
| (c) | Single dilapidated (degraded) property | "Incompatibility with the existing or proposed use; the spatial development framework; and the structure or local plan" |
|  |  | "Diversity of existing use which makes development control difficult or impossible" |
|  |  | "Safety or restriction to the other authorised users" |
|  |  | "High incidence of crime which has been confirmed to be attributable to the type of development" |
| (d) | Uncompleted buildings | "Adverse impact on the environment" |
|  |  | "High incidence of crime which has been confirmed to be attributable to the type of development" |

In the second stage, we used the categorised forms of blight to identify blighted properties within East Legon using a virtual neighbourhood audit technique on the Google Earth aerial image. Neighbourhood audit on the general land use of an area could be reliably conducted with Google street view since the viewer is given a virtual feeling of about 15 m resolution [56]. However, the limitations of this remote observation were the fact that it could only provide the spatial perspective of the blighted properties, which was significantly dependent on the spatial resolution. Furthermore, the coverage was constrained because not all the streets and landed properties in the area could be viewed in the aerial images in 3D. Additionally, Pratomo et al. [57] argue that there are uncertainties regarding the spatial analysis of blighted areas because of non-observable indicators such as land tenure. Additionally, Kohli, Sliuzas, and Stein [58] acknowledge that the accuracy of remote sensing techniques for city deterioration requires some level of tacit knowledge. Thus, we augmented the Google Street view with tacit knowledge of the study area and physical inspections (field investigations).

First, different spots of each category of blight were visually detected on the Google image of the study area using visual image interpretation elements such as pattern, shape, and location/association. According to Bakx et al. [59], pattern depicts the spatial arrange-

ments of the buildings where there is repetition of form, style, or relationships; shape takes into consideration the two or three-dimensional projection of the property with Google Street view; association takes into account the relationship between recognisable features and other structures. In this study, we used the element of shape to identify uncompleted structures within the study area; we also used the association of the blighted property with regards to its surroundings to determine single dilapidated/degraded and uncompleted buildings. Finally, we used pattern and location to determine clusters of disordered settlements and vacant plots, respectively. The four categorised forms of blight are illustrated in Figures 2–5 in the subsequent segment. However, during the field visits, some of the properties initially identified as blight were being developed into ultra-modern structures. This enabled us to further narrow our selection to properties that truly match the different forms of blight, as categorised in Table 2.

3.2.1. The Aerial Views of the Four Forms of Urban Blight

Cluster of Disordered Settlements

Figure 2 illustrates an aerial view of clusters of disordered settlements with blue dots. The selection of this form of urban blight is based on irregularity of plots, overcrowding and lack of access to habitable dwellings.

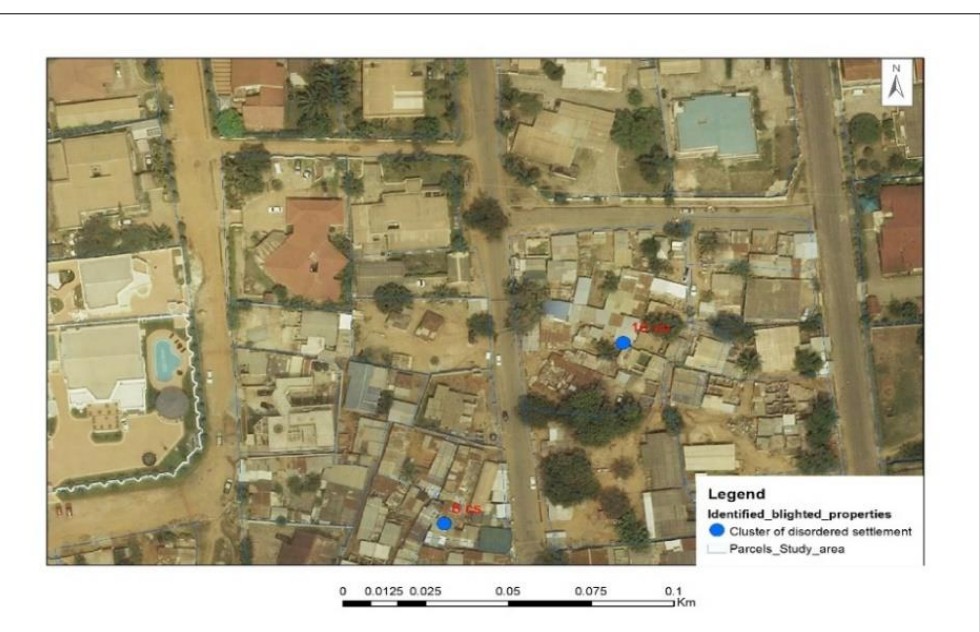

**Figure 2.** Aerial view of cluster of disordered settlements. Source: Google Earth 2018 and parcel plan from Land Use and Spatial Planning Authority.

Vacant Plot of Land

The aerial view of a vacant plot of land surrounded by well-developed properties is shown in Figure 3 with a green dot. The criterion for the selection of vacant plots of land is their undeveloped nature.

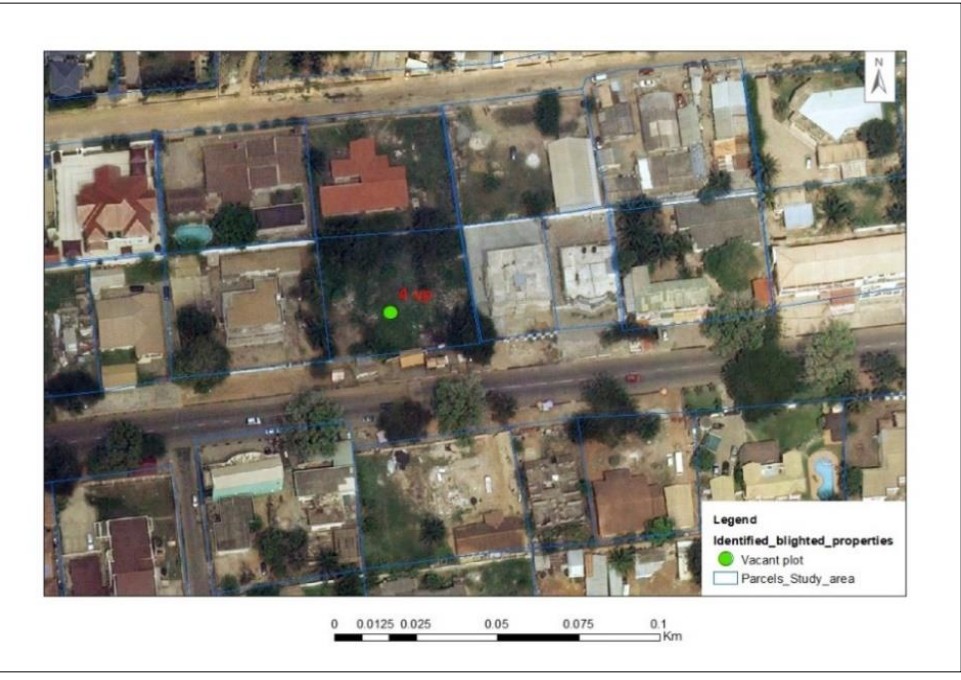

**Figure 3.** Aerial view of a vacant plot of land. Source: Google Earth 2018 and parcel plan from Land Use and Spatial Planning Authority.

Single Dilapidated Property

The selection of dilapidated properties is based on identification as old, obsolete buildings that are degraded or fallen into disrepair. This is shown by the two red dots in Figure 4 below.

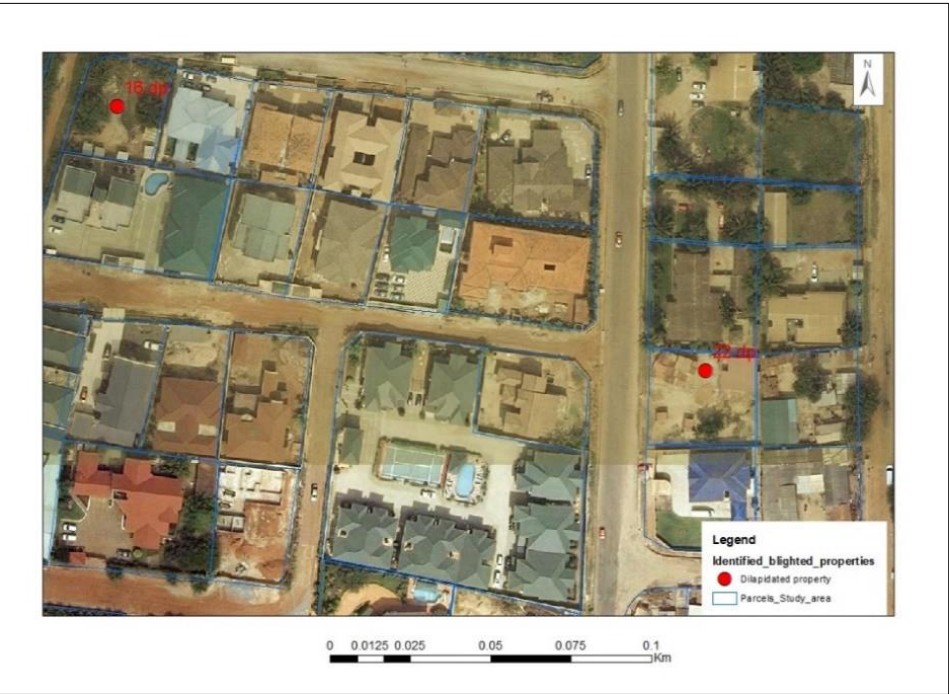

**Figure 4.** Aerial view of dilapidated properties. Source: Google Earth 2018 and parcel plan from Land Use and Spatial Planning Authority.

Uncompleted Structures

Figure 5 shows an aerial view of an uncompleted structure. Aerial selection is based on the foundations of building constructions on site, as shown by the orange dot.

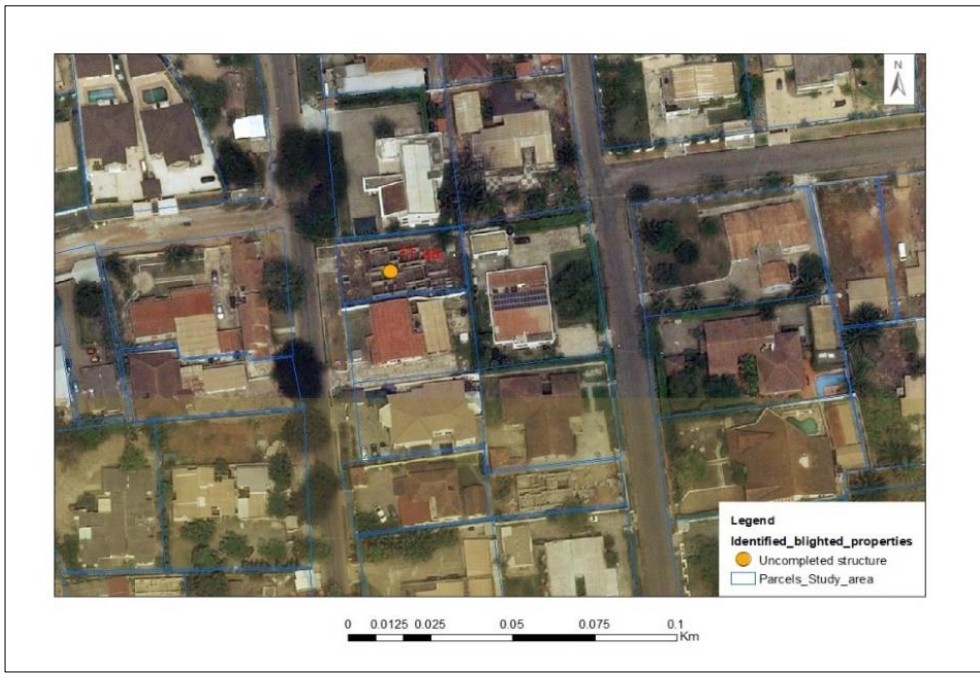

**Figure 5.** Aerial view of an uncompleted structure. Source: Google Earth 2018 and parcel plan from Land Use and Spatial Planning Authority.

*3.3. Data Collection*

Both primary and secondary data were collected for the study. For the primary data, a field investigation was carried out in December 2019/January 2020. This involved purposeful identification of the current distribution of blighted properties. To visually engage and stimulate interest and understanding of urban blight in the respondents, as well as evoke deep reflections, a photograph showing the general description of the mixture of well-developed properties and blighted properties in the study area was used in the interviews (see Appendix B). According to Bryman [60], the photo-elicitation method in qualitative interviews serves as an anchor to trigger, excite and evoke the thoughts, views and perceptions of the respondents to provide a meaningful context for the subject matter of discussion. Furthermore, electronic devices—a digital camera and an audio recorder—were used to capture photographs of blighted properties and conversations with the respondents, respectively. Alternatively, there were narrative recordings and jotting down of notes when respondents were uncomfortable with audio recordings.

Secondary data, on the other hand, were information gathered from scientific articles, journals and aerial images: Google Earth 2018, orthophoto 2016 and the land use plan of the study area obtained from Land Use and Spatial Planning Authority (LUSPA) and Accra Metropolitan Assembly (AMA). The Google Earth aerial image was selected based on spatial resolution to help with the visual image interpretation of the blighted properties. Additionally, the orthophoto gave a better spatial resolution of 0.2 m, as well as spatial data from LUSPA. Furthermore, the boundary of the study area (shape file) was acquired from LUSPA, which was used to locate the study area on the aerial images. The Google Earth image was then exported as a kmz file and subsequently converted to kml files in ArcGIS software and geo-referenced accordingly. Finally, the land use plan assisted with the boundaries of the parcels, which was very helpful to the visual interpretation of the Google Earth image, as shown in Figures 2–5.



### 3.4. Sampling Technique

A study conducted by Galster [37] described the four main actors of neighbourhoods: households, businesses, property owners and local government. In this study, the four key stakeholders considered are experts from statutory agencies, residents/households, property owners and real estate developers. We used non-probability sampling techniques (purposive and convenience sampling) for the selection of respondents. A purposive sampling technique was used to obtain information from the experts. Purposive sampling is the judgment a researcher uses regarding who can provide the needed and required data for a study [61]. In order to understand the dimensions of land tenure and administration, as well as how they feed into the emergence of urban blight, we interviewed four (4) divisional heads of the Greater Accra Regional Lands Commission. Additionally, given that local authorities in conjunction with the Land Use and Spatial Planning Authority (LUSPA) in Ghana have the prerogative of spatial and land use planning, we interviewed four (4) experts from these authorities to determine their perspectives on urban blight, as well as the root causes, characteristics and implementation dynamics that have featured so far in the regulation of property development in East Legon. In this category, we interviewed eight (8) experts for this study.

Additionally, it is important to recognise that urban blight is perceptive in nature and thus may vary across stakeholders/actors. Therefore, we needed to get a clear understanding of how other actors perceive and define urban blight, the socio-cultural practices that surround property holdings, and how such dynamics influence the overall attitude of property management and development within the study context. To find respondents for this category, we used convenience sampling, also known as accidental sampling, which is based on the researcher's ease of accessing, contacting and reaching respondents. Kumar [61] describes convenience sampling as a technique based on suitability and ease of accessing the respondents for a study. Additionally, response saturation (repetition) was used as a guide for our sample size. As explained by Bryman [60], the saturation point is reached by a researcher when either there are no new discoveries of information or any new information is negligible regarding the objective of the study. For this category, we interviewed 22 respondents, which included residents, property owners and real estate developers in the study area. Overall, a total of 30 respondents were interviewed using the two sampling techniques, as presented in Table 3.

**Table 3.** Summary of respondents and sampling strategies.

| No. | Description of Respondents | Sampling Strategy | Total Number |
|---|---|---|---|
| 1 | Residents | Convenience sampling | 12 |
| 2 | Property owners | | 8 |
| 3 | Real estate developers | | 2 |
| | Experts | | |
| 4 | Lands Commission | | |
| i. | Public and Vested Lands Management Division | | 2 |
| ii. | Land Valuation Division | | 1 |
| iii. | Survey and Mapping Division | Purposive sampling | 1 |
| 5 | Land Use and Spatial Planning Authority | | 1 |
| 6 | Local Government Authorities | | |
| i. | Accra Metropolitan Assembly | | 1 |
| ii. | Ayawaso West Municipal Assembly | | 2 |
| | Total number | | 30 |

### 3.5. Data Analysis

The primary data collected via audio recordings were first transcribed into text using Microsoft Word documents. Subsequently, the transcribed documents were uploaded into Atlas.ti software for thematic analysis via open coding in order to identify emerging perceptions and reasons for urban blight. Bryman [60] describes open coding as the process of analysing qualitative data where the researcher remains open-minded to generate as

many ideas as possible as well as make meaning out of the data collected by breaking down, comparing and categorising the data into themes. Additionally, the secondary data obtained from the land-use plan (local plan) and the aerial images were used to generate maps using ArcGIS software, as illustrated in Figures 2–6.

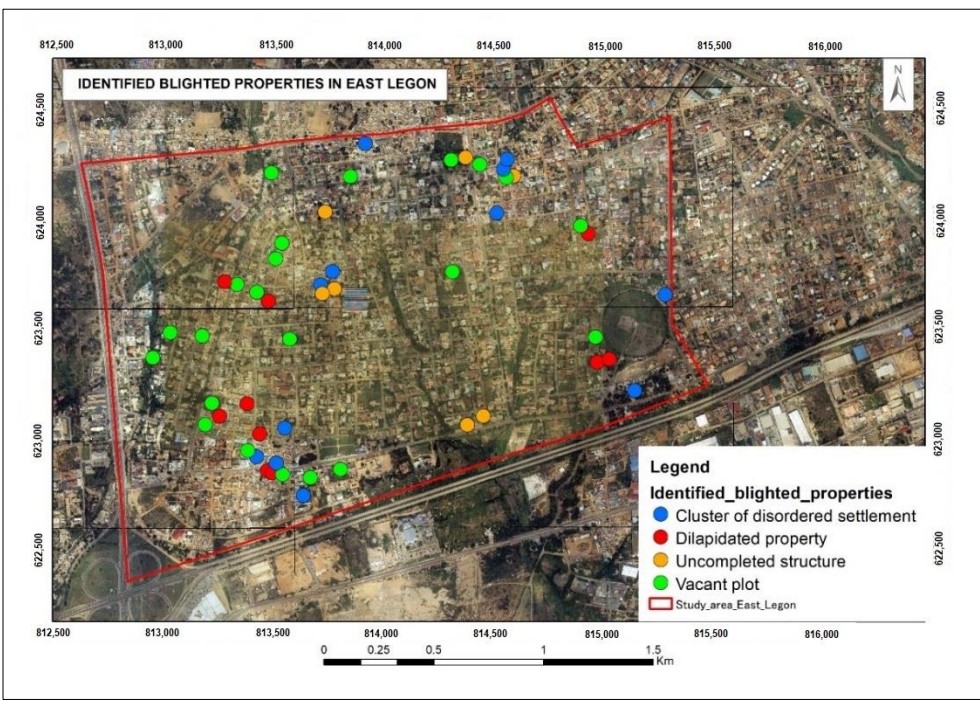

**Figure 6.** Distribution of blighted properties in East Legon. Source: Land Use and Spatial Planning Authority, Orthophoto 2016 and Google image 2018.

## 4. Results

In this section, we present the findings of the study. We begin by illustrating the distribution of urban blight in the study area. In order to ascertain the reasons for blight, we described the perceptions of urban blight across different stakeholders/respondent groups. Next, we present how East Legon became urbanised and subsequently the underlying causes for blight in the area, highlighting how socio-cultural values play a significant role. Finally, we describe the effects of urban blight on the development of the area.

### 4.1. Current Distribution of Urban Blight in East Legon

From the four categorised forms of blight for the study, it was observed that the clusters of disordered settlements are predominantly located at the outer edges of the study area. This confirms the statement made by one of the interviewed experts that planning with infrastructure was carried out in the inner cities whilst the native settlements were left on the outskirts. In terms of frequency (rate of occurrence), vacant/undeveloped plots were most frequent, followed by the clusters of disordered settlements, then dilapidated properties. Uncompleted properties were the least frequent. The observations gathered indicate that some identified uncompleted structures and dilapidated properties were completed or redeveloped into ultra-modern buildings as a result of gentrification. Furthermore, property owners of vacant plots were mostly spectators who were looking for an opportune time to invest in their property. Yet, the cluster of disordered settlements (indigenous and squatter settlements) often stayed in the same condition for many years. The indigenes perceive themselves as natives who must preserve their heritage (customary land rights/properties), whereas squatters (illegal settlers) are mostly immigrants. Figure 6 illustrates the distribution of the forms of blight in the study area.

### 4.2. Local Perception of Urban Blight

Urban blight is a relative concept that is perceived differently by various stakeholders, as demonstrated in Table 4. Generally, urban blight may be regarded as an unsatisfactory use of urban space or real property. Yet, the unsatisfactory manner is usually determined by other stakeholders, while the person using the property may have other values that may be hidden from the other actors. Although urban blight is generally regarded as a negative phenomenon, different respondent groups highlight different perceptions influenced by their relative backgrounds. For instance, drawing from professional standards and intuition, urban planners use non-adherence to land use plans as a reference to establish blight in areas such as squatter settlements. Similarly, residents and property owners in the neighbourhood perceive urban blight from an aesthetic point of view (level of beauty), material culture (the material for construction) and the degree of permanence (squatter settlements). Apparently, most of the respondents perceived squatter (illegal) settlements to be blight, attributing them to the rapid urbanisation. Consequently, there are unauthorised structures portraying non-compliance to city plans and policies. For real estate developers, urban blight is regarded as the non-exploitation of the economic potential of property. The succeeding section provides the background of how East Legon became urbanised.

**Table 4.** Local perception of urban blight.

| | Local Perception | Respondents | | | |
|---|---|---|---|---|---|
| | | Experts | Property Owners | Residents | Real Estate Developers |
| 1 | Aesthetics | | | ✓ | |
| 2 | Indigenous buildings—"*atakpami*" (*local term for mud houses*) | ✓ | | ✓ | |
| 3 | Squatter settlements—"*kiosks*" (*local term for wooden structures built by illegal settlers*) | ✓ | ✓ | ✓ | |
| 4 | Untapped economic potential | | | | ✓ |

### 4.3. Urbanisation in East Legon

Historically, East Legon was a Ga traditional area with four main settlements: Shiashie, Okponglo, Abotsiman, and La Bawaleshie. In 1944, the area was compulsorily acquired by the colonial government as an extension of the International Airport zone. Subsequently, the government carved a portion of the acquired land and created a residential estate for senior civil servants. To all appearances, East Legon became a first-class residential area from the onset after the government established the estate and provided infrastructure such as roads, electricity and water. Information gathered indicates that the indigenes were compensated for the loss of their land inclusive of farmlands. Despite the compensation, they still remained there based on humanitarian grounds. This was articulated by the Land Administrator as "*when the acquisition was done, there was a committee that was set up to look at this particular acquisition. They called it the Alomatu committee. So they were the ones that interacted with the chiefs and the people and they recommended that once they are living there and it is also residential, what is the point in driving them out? They should allow them to be there because, with time, they will grow out of it*". Another expert, Planner, mentioned that urbanisation has caught up with indigenous settlements that were previously at the outskirts of the planned area. He explained, "*unfortunately, developments have sprung up and have made these indigenous buildings to be in the centre of the city but the initial plan had them on the outskirts of the planned area. In order words, the area was planned around them*". Yet, the natives are still glued to their indigenous properties despite being in a first-class area.

*4.4. The Underlying Reasons for Urban Blight in East Legon*

4.4.1. Socio-Cultural Values Attached to Real Property

Socio-cultural values are expressed with deep emotions and intense feelings. They concern traditional and emotional attachments that are not monetary based. It was also established that some properties were named after their ancestors, which had to be preserved for future generations. As it was aptly put by one respondent, "*We have stayed in what our great grandparents built for us. We are also supposed to leave a legacy for generations to come*". Another indigene explained, "*When the property is there, it is immovable . . . anyone who knows the family and wants to trace the family can easily do that because of the property such as somebody who has travelled overseas, he can easily locate the family*". Traditionally, it is believed that real property provides the family lineage and ancestral background of a person. Additionally, people hold on to property as a result of the lasting memories established there. One respondent said, "*This building was put up by our grandmother . . . So we have to preserve her memory for now*". Furthermore, it was recognised that some properties serve as a bolster for life where family members could live, work and go about their daily activities with less stress and traffic than living in the outskirts. As expressed by one respondent; "*they can be in town and transact their activities and businesses, have a place to lay their heads and start life before they go and find their own places . . . development is good but money came to meet human beings. We value human beings and their livelihood, where they will lay their heads, go to work rather than somebody giving you money to take your property*".

4.4.2. Customary Land Control Versus Weak Enforcement of Urban Policies

The spontaneous growth of urban areas without guiding layouts and land use plans has resulted in urban blight. Experts have expressed concerns that not much can be achieved by enforcing urban policies in native settlements termed as "city villages". These indigenes were the customary land owners and first occupants before the formal planning of areas. Invariably, their settlements are often considered as spontaneous growth, although traditional authorities play a critical role in the land use of their jurisdictions. Thus, land use and developments are ahead of planning in most parts of Ghana. Additionally, the study discovered that there are no strict measures for the implementation of urban policies, only eviction and clearing of squatters (illegal settlers). Rapid urbanisation has caused a massive influx of immigrants into the capital city, Accra. Whenever these squatters are evicted from their unplanned dwellings and the cleared space is left undeveloped, the squatters always go back to settle there. According to the expert, there was one such experience in early 2018: "*You know there are recalcitrant people. Even when we cleared them, two (2) to three (3) days later we realized some of them were putting up table tops, containers and kiosks that we had broken down. So whoever is there now, no one has permitted them*". Apparently, after clearing the place in 2018, as at the time of the data collection in December 2019/January 2020, the squatters were still there.

4.4.3. Land Disputes

The land disputes were mainly associated with inadequate transparency regarding land ownerships. Ghana is characterised by a dual system of land tenure: statutory and customary land tenure systems. Most often, conflicting land ownership rights arise among individuals, family members, traditional authorities and government institutions, either between people in the same group or through the involvement of different parties in different groups. Eventually, these conflicts result in litigation, creating a huge backlog of land cases in court, some of which have remained unsettled for many years. The Land Administrator gave an example where the Shiashie family in East Legon instituted a legal case against the Lands Commission for granting leasehold titles to their land. In April 1999, the High Court ruled the case and granted ownership in favour of the Shiashie family. Subsequently, an appeal was made by the Lands Commission in June 2013, and the case was overturned and ruled in favour of the Lands Commission. The latter judgment was based on evidence provided by the Lands Commission regarding the Certificate of Title

for compulsory land acquisition under the Public Lands Ordinance (CAP 34) in 1944. Apparently, for fourteen (14) years of litigation, court orders such as injunctions were placed on land use and development in the areas under contention. Ultimately, blighted properties were left in their blighted conditions, especially the vacant/undeveloped plots and uncompleted structures.

### 4.4.4. Hybrid Land Tenure and Administration

The process and structures involved in managing and disseminating information about the rights and use of land in Ghana are seen to be ineffective by most respondents. Land administration functions cut across both state and non-state (customary) actors. There is a binary distinction in the levels of recording across state and customary land. While customary land tenure system covers about seventy eight percent (78%) of the total land in Ghana, they vary regarding their practices and are mostly undocumented [46,62]. The landholding types under customary land tenure systems in Ghana are family, stool and skin. For the study area, the land ownership is both statutory and customary (family landholding). Due to the inadequate land records, it is difficult to establish certainty of ownership that allows room for fraud and also hinders land transfer. A resident expressed her experience where, in the process of purchasing a blighted property in East Legon, she had to conduct a search at the Lands Commission to ascertain true ownership but never reached an outcome, so she abandoned it, and to date, the property is still in a blighted condition. She stated, "*Land ownership and proof of ownership is difficult in Ghana . . . obtaining information on the ownership alone can take years . . . the whole process is so cumbersome*". Additionally, customary land owners in East Legon (families) are discouraged from regularsing their interest in the land. Explanations for this unwillingness lie in the mechanisms of land rights translation by the Lands Commission. In these processes, the usufructuary rights (superior rights) of families are truncated to leasehold rights, which requires them to pay ground rents (periodic payment of money as a tenant) to the Government.

### 4.4.5. Economic Reasons

Finance is a requisite for property development. While some property owners need capital for the development of their properties, others, such as speculators, are expecting an opportune time to make financial gains from their properties. In the period of waiting, some do not keep their properties in a good state, and they end up becoming blight. Others, on the other hand, choose caretakers or squatters for safety. However, for the indigenous buildings, the question that was asked was, whose responsibility is it to provide capital to develop family property? The indigenous buildings in East Legon are predominantly family properties managed by a family head. By virtue of being a family member, each person has the opportunity to use the property. Nevertheless, regarding development and maintenance, as was stated by one expert, "*The tragedy of the commons happens. Who should connect his resources into redeveloping the family property?*". It was found that some family members with financial capabilities act independently and prefer acquiring personal properties for their nuclear family since any investment made on family land automatically becomes family property per the laws of Ghana.

### 4.5. Effects of Urban Blight on Land Use and Development in the Study Area
### 4.5.1. Positive Effects

Instituting the Historical and Cultural Background of the Area

For the customary landowners, preserving their culture is the main focus. As mentioned earlier, East Legon was hitherto a Ga traditional area. It is believed that most urbanised areas in Accra have indigenous settlements as part of the urban setting. An expert explained, "*When we take every urban settlement, you see the indigenous settlements as part of it . . . you know these villages have always existed*". To all appearances, the indigenous people have settled to maintain their way of living. One resident responded, "*They see

*themselves as they are the owners of the land and they have been there for this long time. It is their property so they take it to be a normal phenomenon so even if you want to develop the place, they won't contribute or participate to make it fruitful. They will just say, after all, we have been here already and we are okay*". Consequently, there is a clear purpose to the present generation holding the land as stewards and leaving a legacy for future generations.

Establishment of Security

For the indigenes and some squatters, the properties, despite their condition, provide security. Security was expressed as a state of well-being and safety. It was discovered that blighted areas are comfortable with less competition. Additionally, the blighted properties keep a low profile and do not expose wealth. Some property owners mentioned that one way to safeguard properties is to keep the exterior of the residence very simple. He stated, "*People look at the exterior of your house to see if there is something valuable. If you look at this building, you will think that there is nothing valuable inside*". Furthermore, some squatters live in blighted properties to save money for future investments. One of them revealed, "*Most of us, not to say we cannot hire a house. For instance, I cannot hire a house but take money say two years advance, Two Thousand, Four Hundred Ghana Cedis (GHC2, 400.00). How much will I use to purchase land? I can use part of the money to buy my own land . . . So it is more like we are also building so we don't want to make more expenses*".

Companionship

Residents in blighted areas feel friendliness among themselves. Some of the respondents expressed that the well-developed properties have fence walls, secured gates with barbed wires, and dogs to deter strangers. Others have security guards and watchmen who would question you whenever you get close to their properties. A resident lamented, "*But what we see is they only think of themselves and fence their property leaving the others*". However, with the blighted areas, there are no such hindrances. They could visit friends without any impediments establishing attachments. An expert explained, "*So in terms of people in these settlements, there is some connection, some attachment and it is difficult to be broken*".

4.5.2. Negative Effects

Aesthetics

Aesthetics concerns the attractiveness of a building. Most of the respondents described well-developed properties and gated communities as pleasant, whereas blighted ones were regarded as being in a poor state. A resident mentioned, "*They don't make the area beautiful . . . but when you get to the gated communities, the buildings alone and the uniformity speak for themselves that it is a beautiful area . . . it makes the area nice and attractive*". Therefore, blighted properties deface the brightness and stunning design of the entire area.

Underutilisation of Properties

Underutilisation was described in terms of the economic potential of a property. This negative effect affects some experts, residents, and real estate developers. The experts mentioned that area classification forms the foundation for property rate. Property rate is described by Asiama [63] as an assessed value of tax levied on real estate properties by the local government authorities. Although East Legon is classified as a first-class residential area, not all properties qualify as such. Thus, it was reported by the experts that the revenue generated from blighted properties is low, which means that the properties are being underutilised despite having high economic potential. Additionally, it was discovered that blighted properties could reduce the value of adjoining properties in a good state. An example given was the sale of a residential property, which, although in a good state, did not command the actual market price as a result of an adjoining blighted property.

Social Inequality and Tension

Inequalities create tension between two groups: those in well-established developments and others. This affects residents and property owners. It was discovered that there is lack of cordial relationships between the two groups. One respondent stated, "*There is no uniformity because you see the clear demarcations between the rich and the poor*". Notably, a master–servant relationship is established between the two groups. Another person responded, "*You realise that most of these people serve as servants for the rich people*". Furthermore, experts and property owners explained the tendency of privacy intrusion; when a well-developed property, like a high-rise structure, adjoins a blighted one which is a single storey, there could be privacy invasion. An expert explained, "*You have done your three bedrooms residential property which is like 30 years old and the next one close to you is a high-rise structure. They are looking into your house and whatever you are doing, they can see*".

Pollution

Air and noise pollution were relatively higher in blighted areas which affects the residents and property owners. This leads to improper waste disposal which contaminates the environment and affects the residents' health. One respondent explained; "*these squatters create a lot of mess around*". Additionally, some vacant plots were considered unkempt with filth where people dump refuse. Furthermore, some residents and property owners complained of the squatters causing noise pollution through loud music and dance in the evenings. A property owner gave an articulate account of his experience; "*where there are squatters . . . . . . . . . .they can really disturb when they play their music. The others come around and dance and have fun. You cannot complain because of democracy and they are also the majority. At times you will be in the room and feel the vibration. This has a negative effect on us*".

Insecurity

Some of the respondents stated that there was some level of insecurity in the area, whereas others compared East Legon to other areas. In comparison, East Legon is more secured. Nevertheless, it was observed that the well-established areas within East Legon are secured. The information gathered indicates that some blighted properties could serve as hotspots for criminal activities. Again, a resident shared his experience: "*In this stretch, there are thieves around. Personally, we experienced theft of African clothes that were put in a locker. Security in this area is low. They can jump from that spot to my place. The rich only secure their places with security men and guards*".

## 5. Discussion

This section reflects and juxtaposes the results of the study with the reviewed literature. Substantial comparison is made between the global north and south. Additionally, the implications of the results of the study on urban land use and policies are considered.

### 5.1. Urbanisation and Distribution of Blight

Urban blight is generally regarded as a negative phenomenon. However, careful reflection on the evolution of urban blight in the global north and south, particularly in the study area, shows an inverse correlation. In the north, a vibrant area was transformed into a deteriorated one [3,5,20,21], whereas in the south, urbanisation transformed an indigenous area and caused the native settlements to be regarded as blight because they do not fit the contemporary setting. Again, rapid urbanisation has resulted in squatter (illegal/unauthorised) settlements due to inadequate housing infrastructure, coupled with lack of social amenities [21]. These relationships are not limited to Ghana but more broadly reflect the character of urban development in Sub-Saharan Africa (SSA) [20,64]. The study found that in the history of urban development in SSA, urban cities were once native settlements, which concurs with the assertions made by Cobbinah and Aboagye [20] and Clarke [65]. Yet, the cities are planned in an ad hoc manner, where infrastructure is provided in certain areas only by neglecting the native settlements. Additionally, the

distribution of blight, as illustrated in Figure 6, confirms this, where clusters of disordered settlements (indigenous and squatter settlements) are predominantly at the outer edges of the study area. Again, in terms of frequency, vacant/undeveloped plots were the most prominent, indicating a high degree of land speculation in the area. Furthermore, uncompleted buildings were the least prominent, and it was observed that some identified blighted properties were being redeveloped. Consequently, the ideas of urban renewal and gentrification have heightened the transformation of existing developments and so-called blighted properties into ultra-modern facilities. These ultra-modern facilities have become the standard of development for existing buildings to be regarded as blight. Careful examination plays an important role in policy development and implementation, with the intention of providing a cosmopolitan community with infrastructure, especially in native settlements. However, there should be strict measures to deter the formation of illegal settlements.

### 5.2. Reasons for Urban Blight

The primary similarity between the cause of blight in both the global north and south is economic factors. However, economic reasons manifest differently. While in the north, economic factors normally manifest on a macro scale, such as the collapse of industries and businesses transforming vibrant cities into abandoned ones, in the south, as in the current study, economic factors largely manifest at the micro level of individual property holders, where, due to lack of finances, properties are not upgraded or left to deteriorate relative to their surroundings. This agrees with the observations made by Crankshaw et al. [17] and Weaver et al. [31] in their studies in South Africa and North England, respectively, suggesting that low-income residents do not maintain a good neighbourhood, which leads to physical stagnation. Another dimension of blight influenced by economic factor is the deterioration of the city landscape resulting from land speculation. Weak land administration coupled with land disputes have caused the property market to be poorly regulated, with some speculators waiting for an opportune time to invest in their property and therefore not keeping their properties in good condition. Instead, they prefer to occupy their land with squatters as a form of security against adverse claim and encroachment. This therefore confirms the argument made by De Soto [66] that property owners in the south are more engrossed in securing their properties by relying on local arrangements.

Additionally, property ownership and/or holdings in the global south are predominantly shrouded in socio-cultural values. As explained by researchers such as Chimhowu [7], Abubakari et al. [8] and Arko-Adjei [46], socio-cultural values have deep roots that not only regulate the interactions between people but also between people and land. The conception of property in many parts of Sub-Saharan Africa (SSA) as a communal entity makes property inseparable from people and also highlights its relevance in establishing one's identity. This resonates with the study of Abubakari et al. [8], which indicates that people trace their identity over time through land. The north, however, is predominantly branded with social values, especially in low-income neighbourhoods where place attachment is established with mutual dependencies on other residents. Additionally, the global north is characterised by few native settlements such as Maori in New Zealand, Aboriginals in Australia, and American and Canadian Indians [42,43]. These northern native settlements are typically concerned with heritage conservation regarding the spirit of their ancestors rather than embedded socio-cultural values in land ownership and use. Apparently, landholdings are predominantly statutory based.

Furthermore, the global south is characterised by fragmented and complex tenure systems. Woven layers of subsisting land rights enable different people or parties (for example, traditional authorities versus government) to lay conflicting claims to land concurrently. The pluralistic land tenure systems provide avenues for encroachment given the weak capacity on the part of planning authorities [20]. A greater percentage of land, specifically in the Sub-Saharan Africa (SSA) area, is under the customary land tenure system, which is largely undocumented, creating forums for fraud and inefficiency of the

property market [7,66]. Additionally, customary landholders are deterred from registering or regularising their interests because of the conversion from freehold to leasehold by the state land agencies. This confirms the claim made by Abubakari et al. [67] that the formal Land Administration System does not recognise the customary freehold, and hence a greater percentage of the property owners are not documented, leading to land disputes. However, in the global north, Zevenbergen [68] highlights that security of tenure is guaranteed either through deeds or land title registration.

*5.3. Effects of Urban Blight*

The main positive effect for both the north and south is place attachment where community bonds and social networks are established [42–44]. Yet, in the south, cultural values are mainly attached to properties [46,49]. Properties are inseparable from people where there is an intended purpose of leaving a legacy for future generations [46]. The intention behind this is that the people (future generations) will not lose their identity [8], thus preserving the cultural background of the area. Additionally, the study found that some squatters rented kiosks in order to save and establish their own residences. This, however, assigns a positive attribute to squatters, which is not frequently found in literature. On the other hand, the predominant negative effect of blight in the north is high criminal activities [14,23], whereas in the south, a dent on aesthetics is the main concern. This confirms the arguments made by Kleinhans et al. [43] and Galster [37] that blighted properties are usually stigmatised when the occupants of blighted properties are less recognised in terms of social status, causing tensions and social imbalances between residents/households.

## 6. Conclusions

Urban blight is a relative concept that is perceived differently by various actors. While urban blight in the global north has been discussed largely from the point of view of its negativity in the broader scheme of city planning and infrastructure development [2,3], the case of East Legon and the global south depict a mixed scenario where urban blight serves a dual function. Whereas urban blight deteriorates the city landscape on a broader level, it also serves the function of promoting cultural heritage for families. More importantly, urban blight in the global south is more nuanced and demonstrates high relativity in what might be considered a blighted property, depending on perspective and purpose. As an emerging trend in developing countries, urban planners and city authorities, in pursuit of city gentrification, promote the development of ultra-modern edifices in native neighbourhoods, which often renders the existing buildings unfit by emerging and contemporary standards. Such induced urban blight not only redefines a criterion for development but, in so doing, opens up cultural homogeneity to heterogeneity and cosmopolitanism. Therefore, in the global south, maintaining a balance between the way of life of a people and emerging global trends of gentrification is essential as the majority of the population are economically vulnerable. It is thus important to identify the significant functions that so-called blighted properties perform within the city core and how such functions strengthen general standards of living and social cohesion. Recognising subtle positive functions in the context of southern cities is crucial in shaping the overall discourse on southern urbanism. The limitation to this study is that the customary land tenure for this study (family landholding type) may not apply to all areas. Customary practices are not a coherent set of stable rules that can be applied uniformly across communities but evolve within and vary across communities [7]. Thus, the influence of land tenure systems on blight may differ because of the pluralistic and varied land tenure systems and practices in the global south. For further studies, the results of this study can be extrapolated to other areas (either within Sub-Saharan Africa or any southern city) with different characteristics, since land tenure systems vary from place to place. Additionally, a study on the role of urban blight and neighbourhood governance in achieving a balance of value systems should be conducted.

**Author Contributions:** Conceptualization, S.A.M., Z.A. and J.M.; methodology, S.A.M.; formal analysis, S.A.M.; investigations, S.A.M.; writing—original draft, S.A.M.; writing—review and editing, S.A.M., Z.A. and J.M.; supervision, Z.A. and J.M. All authors have read and agreed to the published version of the manuscript.

**Funding:** This research received no external funding.

**Institutional Review Board Statement:** Not applicable.

**Informed Consent Statement:** Informed consent was obtained from all subjects involved in the study.

**Data Availability Statement:** Data was archived in University of Twente students database. This is private. However can be accessed with the link:\\UT152200\StudentData/mireku_s6039510.

**Acknowledgments:** To the respondents during fieldwork: the experts for their time and sharing their in-depth knowledge, property owners and residents in East Legon.

**Conflicts of Interest:** The authors declare no conflict of interest.

## Appendix A  Analytical Framework

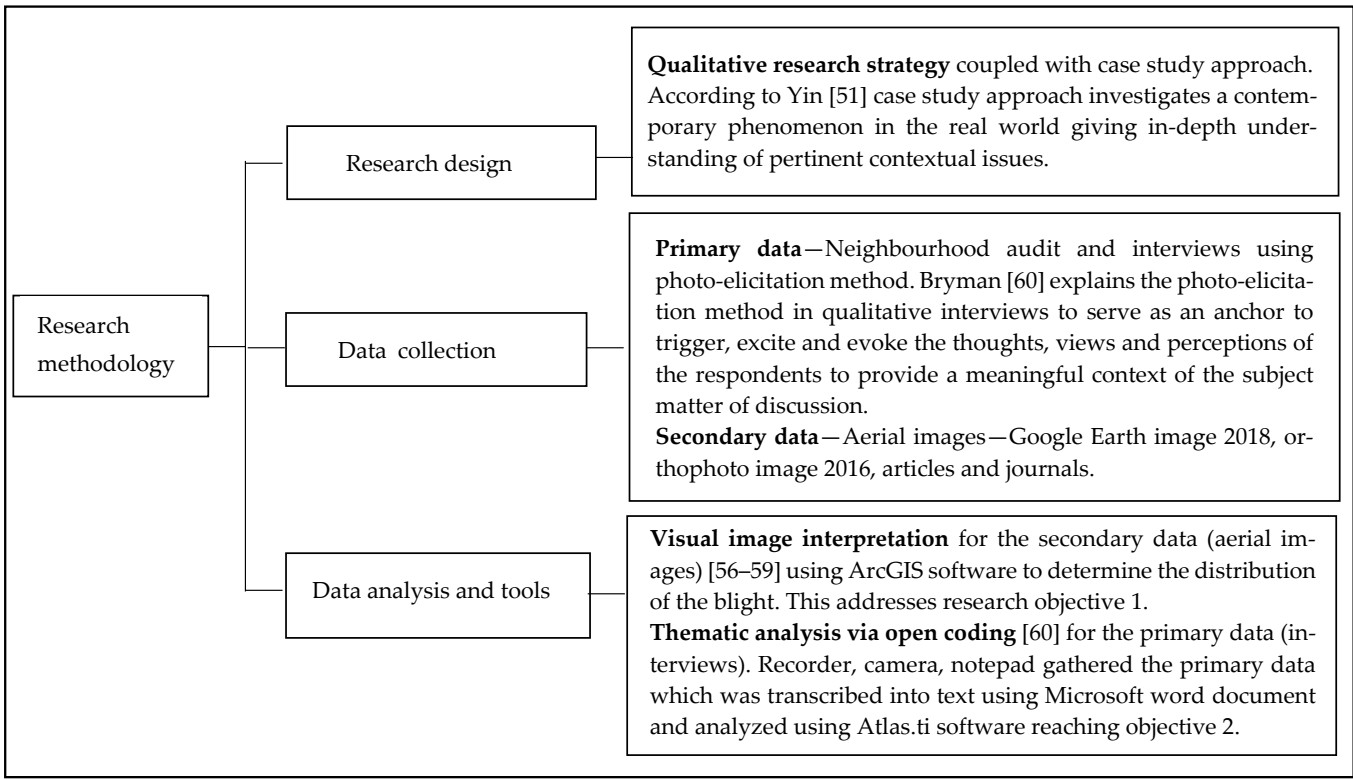

**Figure A1.** Analytical framework showing the research steps for the study.

**Appendix B  Photograph Used for Photo-Elicitation during Fieldwork**

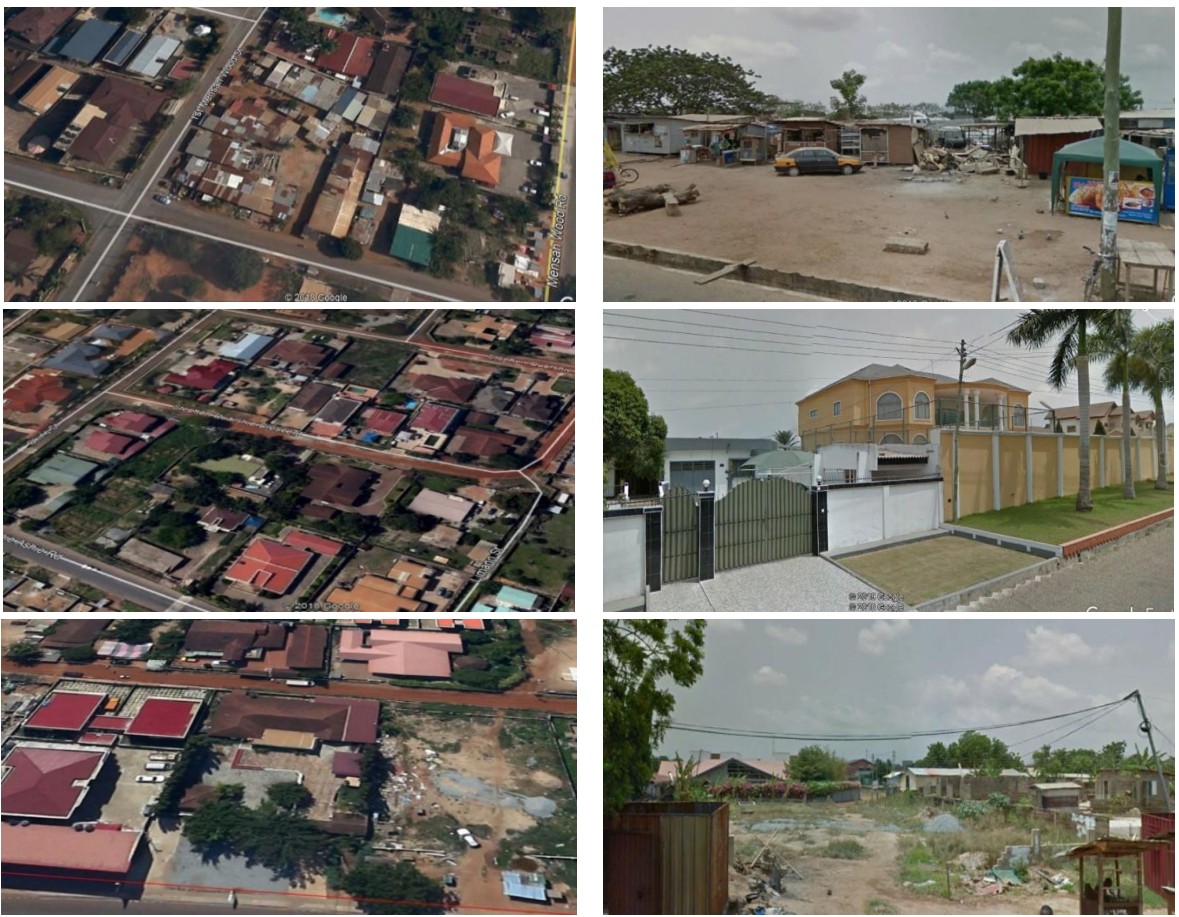

**Figure A2.** Views of both blighted and well-developed properties.

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
