# Peer review of "Dimensions of Urban Blight in Emerging Southern Cities: A Case Study of Accra-Ghana"

_sustainability, doi:10.3390/su13158399_

Round 1
Reviewer 1 Report
Dear author,
The paper is interesting and original, as it provides insights on dimensions of urban blight in emerging southern cities.
Below I would give some comments to better clarify the research path and structure in line with the MDPI template (https://www.mdpi.com/journal/sustainability/instructions):
- Introduction
The theoretical background is suitable for the aim of the paper, but I would suggest implementing the literature review on socio-cultural values in urban regeneration processes. I would also suggest merging this with paragraph 2 that describes key references on urban blight. Please, at the end of the introduction in briefly mention the paper Sections for highlighting the research structure in responding to the research question.
- Materials and methods
Materials and methods could be better explained, I would suggest inserting a research methodology explanation specifying each research step able in responding to the research objectives (a graphic on research methodology approach could help). I would also suggest better highlight which research analytical framework is used for the case study analysis and which qualitative methods used. For every research step, please show tools and approaches used with literature references.
- Results
Please, explain for every research step the results achieved.
- Discussion and conclusions
The findings and their implications should be discussed also with limitations of the work highlighted. The paragraph of discussion could be merged with conclusions and I would suggest implementing the discussion for each research step.
In general, within the whole paper paragraphs, I would avoid dividing the paragraph into short subparagraphs.
Author Response
Dear reviewer,
Thank you for your review. Kindly find attached the responses to the comments in the word document.

Reviewer 2 Report
- It is and interesting study, potentially applicable in similar conditions not only in Africa, provided that the methodology will be progressively migrating to a quantitative approach.
- Actually, the methodology to recognize the blight areas (abandoned, fragmented, incomplete, etc.) is affected by relevant subjectivity, though the sources (google maps, etc.) are normally available and good quality.
- Besides, the integration with interviews is reasonable but should be built in a more structured way, starting from reliable demographic and socio-economics data. This aspect is not detailed at all.
- The sampling of 30 respondents is chosen without any representativeness criteria analysis. This must be clarified.
- The required knowledge of local areas makes difficult to replicate the method in various context without involving the local society. This should be considered.
- The situations with squatters, land disputes, etc. are detected, without and explanation about how. Besides, the methodology does not explain potential solutions to apply in these situations.
- The detection of critical situations and their causes and effects are mainly dependent on the interpretation of written sentences or interviews, which makes it almost impossible to generalize the methodology.
Author Response
Dear reviewer,
Thank you for the review. Kindly find attached the responses to the comments made.

Reviewer 3 Report
This is a relatively good and interesting paper, well illustrated and documented. However, I would see some revisions before acceptance, as follows:
- Literature review is partial and, in some instances, incomplete. Please enrich this part in a truly international perspective.
- Degradation of aesthetic beauty and landscape are important issues in cities, that should be theoretized more clearly and esplicitly in the paper.
- Discussion is really short and not conclusive, rather disconnected with the results, and very technical, being also organized in short paragraphs. I would see a more discorsive discussion with a broader reference to earlier studies.
- Language usage needs some clarifications, including use of terms (e.g. are you sure 'dilapidation' is the right term?).
Author Response
Dear reviewer,
Thank you for your comments. Kindly find attached the responses to the comments made.

Round 2
Reviewer 2 Report
Thanks to the authors for taking into consideration my suggestions. I think the paper is now substantially improved.
Reviewer 3 Report
Good paper and rather good revisions overall. Only minor language checks (customary, MPDI) are necessary! Thank you